# Bioaccumulation of Lead, Cadmium, and Arsenic in a Mining Area and Its Associated Health Effects

**DOI:** 10.3390/toxics11060519

**Published:** 2023-06-09

**Authors:** Ulziikhishig Surenbaatar, Seungho Lee, Jung-Yeon Kwon, Hyunju Lim, Jeong-Jin Kim, Young-Hun Kim, Young-Seoub Hong

**Affiliations:** 1Department of Preventive Medicine, College of Medicine, Dong-A University, Busan 49201, Republic of Korea; ulzii1024@dau.ac.kr (U.S.); kitty1004ki@dau.ac.kr (J.-Y.K.); yshong@dau.ac.kr (Y.-S.H.); 2Environmental Health Center, Dong-A University, Busan 49201, Republic of Korea; hyounli@dau.ac.kr; 3Department Civil-Environmental Engineering, Andong National University, Andong 36729, Republic of Korea; jjkim@andong.ac.kr (J.-J.K.); youngkim@andong.ac.kr (Y.-H.K.)

**Keywords:** mine, bio-concentration factor, biomonitoring, heavy metal, estimated glomerular filtration rate

## Abstract

Soil contamination is associated with a high potential for health issues. This study aimed to investigate the bioaccumulation of heavy metals and its associated health impact among residents near a mining area. We performed environmental monitoring by analyzing lead (Pb), cadmium (Cd), and arsenic (As) levels in soil and rice samples, as well as biomonitoring by analyzing blood and urine samples from 58 residents living near the mine. Additionally, concentration trends were investigated among 26 participants in a 2013 study. The Cd and As levels in the soil samples and Cd levels in the rice samples exceeded the criteria for concern. The geometric mean blood Cd level (2.12 μg/L) was two times higher than that in the general population aged > 40 years. The blood Cd level showed decreasing trends from the previous measurements of 4.56–2.25 μg/L, but was still higher than that in the general population. The blood and urine Cd levels were higher in those with a low estimated glomerular filtration rate (eGFR) than in those with normal eGFR. In conclusion, heavy metals from mining areas can accumulate in soil and rice, adversely impacting human health. Continuous environmental monitoring and biomonitoring are required to ensure the safety of residents.

## 1. Introduction

Some heavy metals occur naturally in the air, water, and soil. Soil has low mobility; thus, soil contamination leads to continuous exposure to, and accumulation of, agricultural products. Ingesting agricultural products and water contaminated with Cd causes Itai-itai disease, a representative heavy metal-related disease that leads to bone decalcification. Exposure to Pb, Cd, and As is also associated with an increased risk of cardiovascular disease and coronary heart disease [1]. Previous studies reported that co-exposure to Pb and Cd can be a strong determinant of chronic kidney disease [2]. In addition, chronic exposure to arsenic is a risk factor for lung, bladder, and skin cancers [3]. As, Pb, and Cd are included in The US Agency for Toxic Substances and Disease Registry’s priority list of 275 hazardous substances, where they are listed as items 1, 2, and 7, respectively [4].

Pb, Cd, and As do not have a metabolic function in the human body [5], but they are commonly absorbed via inhalation or ingestion, with the degree of absorption varying depending on the route of exposure. Once inhaled or ingested, they are primarily transported via the blood and accumulate in other organs and tissues. The half-life of Pb in the blood is approximately 28–36 days. However, approximately 90% of Pb is distributed in the bone, where it has a half-life of more than 20 years. [6]. Meanwhile, Cd binds to metallothionein in the liver after exposure and accumulates in the kidney, with a half-life of approximately 6–38 years [7]. As it accumulates in various tissues, inorganic As is more toxic than organic As. The half-life of inorganic As in the blood is 4–6 h, while that of organic As is as long as 20–30 h [8].

Following the National Health and Nutrition Examination Survey [9] and the Canadian Health Measures Survey (CHMS) [10] recommendation, many countries have implemented biomonitoring surveys to assess the levels of exposure to heavy metals and other hazardous substances among their citizens. In Korea, two nationwide biomonitoring surveys are performed: the Korean National Health and Nutrition Examination Survey (KNHANES), conducted by the Korea Centers for Disease Control and Prevention, and the Korean National Environmental Health Survey (KoNEHS), conducted by the Korea National Institute of Environmental Research. The KoNEHS provides representative values for the general population by measuring environmental contaminants in biological samples and the factors influencing the levels of these contaminants every 3 years.

As of 2020, 2428 abandoned mines have been identified in Korea [11]. In 1995–1996, some residents near the Gahak mine were exposed to heavy metals through crops grown in contaminated soil, and in 2004, Itai-itai disease was recorded in another mining area [12,13,14]. Accordingly, comprehensive analyses of soil and agricultural products are necessary for determining the overall level of environmental contamination and assessing the risk posed to residents living near mining areas. However, exposure assessments are usually performed using environmental or biological samples. Thus, a comprehensive study that involves simultaneous environmental monitoring and biomonitoring is needed. 

The present study aimed to investigate the bioaccumulation of heavy metals and their associated health impact among residents living near a mining area. To this end, we performed an integrated assessment of the biomonitoring findings and environmental sampling near a mining area.

## 2. Materials and Methods

### 2.1. Participant Recruitment and Sample Collection

The Korean Ministry of Environment had previously conducted a pilot study of a risk assessment among residents living near mines that have been abandoned since 2012. Among these mines, the Jinheung Hongcheon mine, located in Gyeongsang province, was chosen for the detailed re-investigation in 2021 because of the high concentrations of heavy metals found in the environment and residents. The area was investigated by collecting topsoil (0–30 cm) from 8 agricultural fields (S1–S8) and 2 forests (S9, S10) between the mine shaft and the village. All sampling points were located within a 2 km radius of the mine pit. Eight grain samples (G1–G8) were collected from rice grown at the points of the soil sampling.

This study enrolled individuals who lived in the mining area for >5 years. Among the 68 residents aged >18 years who initially participated, 10 residents were excluded because they had lived in this area for <5 years. Finally, a total of 58 residents participated in the study, including 26 subjects who participated in the 2013 survey. Face-to-face interviews were conducted to obtain data on demographic and socioeconomic characteristics, such as occupational history, smoking, drinking, history of chronic disease, and dietary habits. All information was obtained by trained interviewers using a standardized questionnaire. Whole blood (at least 5 mL) was collected and stored in ethylene diamine tetra acetic acid (EDTA)-coated vacutainers. All blood samples were stored at −70 °C until analysis. Urine samples were also collected by providing the participants with sample containers and instructing them to directly collect approximately 30 mL of spot urine. The urine samples were then divided into two conical tubes and stored at −70 ℃ until analysis. This study was approved by the institutional review board at Chung-Ang University (IRB No. 1041078-202103-BRHR-076-01), and each participant provided written informed consent. The study workflow is illustrated in Figure 1.

### 2.2. Sample Preparation and Instrumental Analysis

After natural drying, 3.0 g soil samples were passed through a 100-mesh sieve, weighed, placed in test tubes containing 60% nitric acid (7.0 mL) and 35% hydrochloric acid (21 mL), and heated for 1 h at 70 °C on a heating block. Pb, Cd, and As levels in the obtained samples were analyzed using an inductively coupled plasma-optical emission spectrometer (ICP-OES; Agilent 720; Agilent Technologies, Santa Clara, CA, USA). The peeled rice samples were dried in an oven at 40 °C for more than 24 h and then finely pulverized using a mortar. After placing a 1.0 g sample with 10 mL of nitric acid into a microwave digestion vessel (UltraWAVE; Milestone, Brøndby, Denmark), an additional 5 mL of 65% concentrated nitric acid was added. The decomposition vessels were left in a ventilation hood for 1 h to remove the generated gas. Following the sufficient removal of the gas, the microwave digestion vessels were placed in a microwave device, and the samples were decomposed at 230 °C and 40 bar for 30 min. To lower the acidity of the samples, the decomposition solutions were diluted with distilled water (38–40 mL) in 50 mL conical tubes. The obtained diluted solutions were analyzed for the Pb, Cd, and As levels using an inductively coupled plasma-mass spectrometer (ICP-MS; Agilent 7800; Agilent Technologies, Santa Clara, CA, USA).

The Pb and Cd levels in the blood were measured using an ICP-MS (NexION200B; PerkinElmer, Waltham, MA, USA). A calibration curve using a mixed standard solution was prepared by diluting Pb and Cd standard solutions in 1% nitric acid (HNO_3_). To prepare solutions at each of the 7 concentrations, 0.1 mL of the standard sample was added to the diluting solution to achieve a total volume of 5 mL. The samples were analyzed using the standard addition method. Ammonium pyrimidine dithiocarbamate (0.01%), ethanol (1%), tetramethyl ammonium hydroxide (25%), Triton X-100 (0.05%), and nitric acid were added to tertiary distilled water to prepare the diluting solution. In 4.8 mL of this solution, 0.1 mL of 1% HNO_3_ and 0.1 mL of blood sample were thoroughly mixed for complete hemolysis, and the centrifuged supernatant was used for analysis. Iridium at 1000 mg/L (SPEX CertiPrep, Metuchen, NJ, USA) was used as an internal standard. Blood ClinCheck levels 1, 2 (RECIPE Chemicals, Munich, Germany) were used as the certified reference materials for internal quality assurance and management [15].

The Cd levels in the urine samples were measured using an ICP-MS (Agilent 7700; Agilent Technologies, Santa Clara, CA, USA). A standard solution of Cd diluted in 2% 1-butanol, 0.05% EDTA, 0.05% Triton X-100, 1% NH_4_OH, and tertiary distilled water was used to prepare the calibration curve. Standard samples were collected and mixed with the pooled human urine filtered (Innovative Research Inc., Novi, MI, USA). The resulting mixture was diluted to 0.05, 0.1, 0.5, 1, and 2 μg/L using a diluting solution. The diluted solutions were then analyzed using the standard substance addition method. The urine sample was diluted using the diluting solution in a 1:10 ratio and thoroughly mixed to be used for analysis. Rhodium at 10 mg/L (Agilent Technologies, Santa Clara, CA, USA) was used as an internal standard. Urine ClinCheck levels 1, 2 (RECIPE Chemicals, Munich, Germany) were used as the CRM for internal quality assurance and management.

For the isolation and quantification of As species, the urine samples were diluted 10-fold with deionized water after removing impurities through a 0.45 µm filter. Four As species (As^3+^, As^5+^, monomethylarsonic acid, and dimethylarsinic acid) were quantitatively analyzed using a liquid chromatography (LC)/ICP-MS device (High-Performance Liquid Chromatography, Agilent 1260; Agilent Technologies/ICP-MS, Agilent 7700, Agilent Technologies) equipped with a Hamilton PRP X-100 column. The accuracy of the calibration curves was verified using two standard materials (Standard Reference Material 2669, The National Institute of Standards and Technology, Gaithersburg, MD, USA; Certified Reference Material No. 18, National Institute for Environmental Studies, Onogawa, Tsukuba, Japan).

The external quality assurance process for each heavy metal analysis was accredited by the Quality Assurance Program of the German External Quality Assessment Scheme operated by Friedrich-Alexander University.

### 2.3. Bio-Concentration Factor

The bio-concentration factor (BCF) indicates the degree of transfer from soil to crops [16]. In this study, the BCF was defined as the ratio of the concentration of heavy metals accumulated in crops to their concentration in the soil and was calculated using the following equation based on the heavy metal levels measured in farmland soils and rice: $$BCF=\frac{M_{crop}\left(mg/kg,dryweight\right)}{M_{soil}\left(mg/kg,dryweight\right)}$$

### 2.4. Calculation of Estimated Glomerular Filtration Rate

The estimated glomerular filtration rate (eGFR) was calculated using the Chronic Kidney Disease-Epidemiology Collaboration creatinine equation, as follows [17]: (i) serum creatinine level ≤ 0.9 mg/dL and male sex: eGFR = 142 × (serum creatinine/0.9)^−0.302^ × (0.9938)^Age^); (ii) serum creatinine level > 0.9 mg/dL and male sex: eGFR = 142 × (serum creatinine/0.9)^−1.200^ × (0.9938)^Age^; (iii) serum creatinine level ≤ 0.7 mg/dL and female sex: eGFR = 144 × (serum creatinine/0.7)^−0.241^ × (0.9938)^Age^; and (iv) serum creatinine level > 0.7 mg/dL and female sex: eGFR =144 × (serum creatinine/0.7)^−1.200^ × (0.9938)^Age^. The eGFR values were classified into three categories (low, <60 mL/min/1.73 m^2^; intermediate, 60–89 mL/min/1.73 m^2^; and normal, ≥90 mL/min/1.73 m^2^) based on the Kidney Disease: Improving Global Outcomes guidelines [18].

### 2.5. Statistical Analysis

The distributions of the general characteristics, morbidity (e.g., blood pressure and diabetes), and heavy metal levels were initially analyzed to identify independent variables. Given that the distributions of the biomonitoring levels were right-skewed, the levels were log-transformed prior to all statistical analyses. Changes in the blood Pb and Cd levels between 2013 and 2021 were analyzed for the 26 participants with the data for both years. In addition, the geometric means (GMs) for the general population were calculated using the data from the 2017 KNHANES and the 3rd KoNEHS (2015–2017). Considering that the KNHANES provides the blood Cd levels, whereas the KoNEHS provides the urinary Cd levels, the GMs were calculated for individuals aged > 40 years using each dataset. The reference value (RV95) for the general population was set as the guidance value.

Next, the GMs of the blood and urine levels of Cd were calculated for each eGFR category. Multiple logistic regression analyses adjusted for age, sex, duration of residence, and levels of Pb and As were performed to analyze the association between the Cd levels and eGFR. All statistical analyses were performed using STATA (Ver 17.0; Lakeway Drive, College Station, TX, USA). Statistical significance was set at a 5% level.

## 3. Results

### 3.1. Heavy Metal Concentrations in Environmental Samples

The Pb levels in the soil samples from the farmland ranged between 12.64 mg/kg (S5) and 25.04 mg/kg (S1), which were within the normal limits. Similarly, the Cd levels did not exceed the criterion for concern (4 mg/kg), with a mean of 1.16 ± 0.43 mg/kg and a range of 0.62 (S8) mg/kg to 1.80 (S3) mg/kg. Meanwhile, the Cd levels in the samples from point S9, located in front of the mine shaft, exceeded the criterion for concern for the forest area (10 mg/kg), with a measured level of 11.09 mg/kg. The mean As level in the farmland samples was 23.77 ± 10.80 mg/kg (range: 8.38 (S8) mg/kg to 38.18 (S5) mg/kg). The As levels of the samples from four points (S2–S5) exceeded the criterion for concern (25 mg/kg). In addition, the As level from the forest sample (S10) also exceeded the criterion for concern (150 mg/kg) [19].

The criterion for concern for Pb, Cd, and As levels in rice is 0.2 mg/kg [20]. In this study, the mean Cd level in the rice samples was 0.36 ± 0.56 mg/kg. The Cd levels in the samples from S2 and S5 exceeded the criterion of concern, whereas the Pb and As levels did not (Table 1). The mean BCF of Pb, Cd, and As was 0.007 (range: 0.00–0.029), 0.252 (range: 0.014–1.06), and 0.003 (range: 0.0014–0.005), respectively. The BCF was the highest for Cd. None of the Pb or As in any of the samples exceeded a BCF of 1.0; however, the BCF of Cd was over 1.0 in the samples from S2 and G2, suggesting that Cd absorption is greater in rice than in soil (Figure 2).

### 3.2. General Characteristics of the Study Participants and Concentrations of Heavy Metals in Biological Samples

The average age of the 58 participants was 69.4 years. Among them, 19 and 39 were male and female, respectively, and the average age was 67.9 ± 10.2 and 70.1 ± 12.7 years, respectively. The average body mass index was 23.8 ± 3.9 kg/m^2^_,_ and no sex differences were observed. The average duration of residence was 45.3 years. There were two participants who had previously worked in mines (one male and one female). The proportions of smokers and drinkers were higher in men than in women. Overall, 40 participants (69%) reported that they had eaten rice grown near the abandoned metal mine. Hypertension and diabetes were diagnosed in 25 (44.5%) and 9 (15.5%) participants, respectively (Table 2).

The changes in the blood Pb and Cd levels among the 26 participants are shown in Figure 3. Each line was sorted by the measurement in 2013 and then expressed the trend of the measurement in 2021. In general, the blood levels of Pb decreased among those who had high Pb levels in 2013. However, in one participant with low Pb levels in 2013, the levels increased from 2.32 μg/dL to 7.53 μg/dL. The GM of the blood Pb levels for the local residents was 1.80 μg/dL (95% confidence interval [CI]: 1.54, 2.09), similar to the blood Pb level of 1.68 μg/dL (95% CI: 1.64, 1.71) reported for those aged > 40 years in the KNHANES (Table 2). Meanwhile, the blood levels of Cd tended to decrease, from 4.56 μg/L in 2013 to 2.25 μg/L in 2021 (Figure 2), although the level is still higher than that of the general population (Figure 4A). The GM of the urinary Cd levels was 2.69 μg/L (95% CI: 2.23, 3.25), higher than the 95th percentile (2.13 μg /L) from the KoNEHS (Figure 4B). In all participants, the GM of the blood Cd level was 2.12 μg/L (95% CI: 1.73, 2.59), approximately twice as high as the GM (1.02 μg/L) in those aged > 40 years in the KNHANES (Table 2). The GMs of the urinary inorganic As and total As levels were 0.22 μg/L (95% CI: 0.14, 0.33) and 51.8 μg/L (95% CI: 42.4, 63.2), respectively. Comparison with the nationwide monitoring data was not performed because urine As was not measured.

### 3.3. Correlation between Cd Levels and Renal Function

The low eGFR group was significantly older than the normal and intermediate eGFR groups. Two participants with low eGFR were diagnosed with diabetes (Table 3). The average blood Cd levels were 7.74 μg/L in the low eGFR group, 2.15 μg/L in the intermediate eGFR group, and 1.99 μg/L in the normal eGFR group. Although the blood Cd levels were higher in the low eGFR group than in the normal eGFR group, the difference was only marginally significant (*p* = 0.076). The urinary Cd levels were 3.66, 3.88, and 2.48 μg/L in the low, intermediate, and normal eGFR groups, respectively. Although the urinary Cd levels were higher in the low and intermediate eGFR groups than in the normal eGFR group, the differences were not statistically significant.

## 4. Discussion

In this study, we performed a comprehensive analysis of heavy metal exposure among residents near an abandoned metal mine based on environmental monitoring and biomonitoring assessments. Our environmental monitoring results indicated that the Cd and As levels in the soil samples exceeded the criteria for concern. In addition, the Cd levels in the rice samples were close to the criteria for concern and had a BCF exceeding 1.0 in some samples. The blood Cd levels were two times higher and the urinary Cd levels were four times higher than those in the general population. Furthermore, the Cd level was approximately three times higher in the low eGFR group than in the normal group.

The mean Pb level in the soil samples (15.51 mg/kg) was lower than that reported in the national survey of heavy metal contamination conducted in 2020 (18.89 mg/kg), whereas the mean Cd (1.16 mg/kg vs. 0.121 mg/kg) and As levels (23.77 mg/kg vs. 5.45 mg/kg) were considerably higher [21]. In a study by Jang et al., the GMs of Pb, Cd, and As in areas near abandoned mines were 8.61, 0.186, and 1.81 mg/kg, respectively [22]. Compared with the values reported in 2014 (soil: 1.29 mg/kg, rice: 0.14 mg/kg) [23], the levels of Cd in the soil found in the current study were lower, but those in the rice samples obtained from the same area were higher (soil: 1.16 mg/kg, rice: 0.36 mg/kg). The relatively higher levels observed in the current study may be explained by differences in the sampling time and location. In addition, our analyses indicated that the concentrations of heavy metals in the soil decreased over time, while the concentrations in crops increased over time.

The bio-concentration of heavy metals in agricultural products is the main source of human exposure, and the chronic accumulation of heavy metals is a known health hazard [24]. The BCFs of Cd (GM: 0.089, AM: 0.252) and As (GM: 0.002, AM: 0.003) in the crop samples from this study were lower than those reported by the US Environmental Protection Agency (USEPA) (Cd, GM: 0.36; As, GM: 0.026) [25]. However, the levels in some individual rice samples exceeded the average values reported by the USEPA (BCF for S2/G2: 1.06). Kim et al. analyzed rice samples from the farmland and found that the BCFs of Pb, Cd, and As were 0.051, 0.019, and 0.005, respectively [26]. Thus, the BCF of Cd (0.252) was relatively high in the current study. In another study, the BCF of Cd in rice from farmland near a mine was 0.018 [27], further suggesting that the BCF of Cd in this study was high. These discrepancies may be explained by the differences in the total heavy metal content and phyto-availability, as well as by physiological differences between crops. Lim et al., reported high concentrations of Cd and Pb in rice, soybean, sesame, and corn, in this order [27]. The hazards associated with contaminated agricultural products are proportional to the intake level. Therefore, individuals with a greater consumption of contaminated crops are at higher risk. Rice is a staple food in Korea; thus, it is the major source of Cd exposure. Therefore, regular environmental monitoring of soil and water samples is necessary to ensure the safety of agricultural products [28].

The GM of the blood Cd among the participants was 2.12 μg/L, approximately two times higher than the GM for those aged > 40 years (1.02 μg/L) in the general population. The GM of the urinary Cd was 2.96 μg/L (3.52 μg/g creatinine), approximately four times higher than that for the general population. National reports from the US and Canada showed that the GMs of blood Cd and urinary Cd were 0.30 μg/L and 0.18 μg/L (NHANES 2015–2016) and 0.24 μg/L and 0.19 μg/L (CHMS 2018–2019) [9,10], respectively. Although the levels were high among those residing near abandoned mines, the mean blood levels of Pb (2.44 μg/dL→1.71 μg/dL) and Cd (4.56 μg/L→2.25 μg/L) tended to decrease among the 26 participants who had also participated in the 2013 study. 

Ahn et al. reported a mean blood Cd level of 5.33 μg/L and a mean urinary Cd level of 5.31 μg/L among individuals residing near abandoned mines, which differed significantly from the levels observed in the control area (blood Cd: 1.63 μg/L, urinary Cd: 1.02 μg/L) [23]. Another study reported that the GMs of the blood Cd and urinary Cd levels for those living near abandoned mines were 2.92 μg/L and 1.53 μg/g creatinine, respectively [29]. Interestingly, the blood Cd level in this study (2.12 μg/L) was lower, whereas the urine Cd level (3.52 μg/g creatinine) was more than two times higher. Chung et al. reported that the high urinary Cd level (2.79 μg/g creatinine) of some residents near mines may be attributable to crop contamination [30]. Similarly, the current study found that the Cd levels were significantly higher among residents near abandoned metal mines than among those in the control area, and this was likely due to heavy metal exposure from the abandoned metal mines. Given the high Cd levels in the soil and rice samples, the high Cd levels may have resulted from the consumption of crops grown in the area.

As an indicator of Cd exposure, blood Cd levels reflect exposure over a span of recent weeks to several months, while urinary Cd levels represent the total accumulation in the blood and kidneys [31,32]. In this study, the blood Cd levels exhibited a weak correlation with the urinary Cd levels (r = 0.338, *p* = 0.009) (data not shown). The biological half-life of Cd is up to 38 years, and it accumulates in the body over one’s lifetime, with the main target organ being the kidney [33]. Thus, chronic exposure to Cd, even in small amounts, can lead to its accumulation in the kidney and the deterioration of glomerular function over time. Bulter et al. reported a negative association between the blood Cd levels and eGFR (β: −4.23, *p*-value: <0.05) in sugarcane workers [34]. Another study reported that the blood Cd (median: 0.38 μg/L) and urinary Cd (median: 0.52 μg/L, AM: 0.6 μg/L) levels of Swedish women influenced glomerular function, despite their relatively low levels [35]. Moreover, impairments in renal function due to high levels of exposure may not resolve after decreases in Cd exposure [36]. Liang et al., found a significant decrease in Cd levels (blood Cd: 8.9 μg/L→3.3 μg/L, urinary Cd: 11.6 μg/L→9.0 μg/L) with increasing urinary albumin levels after reducing Cd exposure, but renal tubular dysfunction did not fully recover [37]. Furthermore, acute exposure to Cd temporarily increases the renal concentration of Cd, which may result in acute renal toxicity [38,39]. 

In this study, 18.9% (*n* = 11) of the participants had lower than normal eGFRs, and marginally significant differences in the blood Cd levels were observed among the three eGFR groups. Although the difference was not significant, the urinary Cd level was markedly lower in the participants with an eGFR ≥ 90 than in those with an eGFR < 90. Two patients with eGFRs < 60 were diagnosed with diabetes. Diabetes is closely related to eGFR. However, it was difficult to determine whether the Cd was not filtered and had accumulated in the body due to diabetes or the decreased eGFR resulted from Cd exposure. The mean age was also higher in the low eGFR group than in the normal eGFR group. Our multiple logistic regression analyses indicated that age significantly influenced eGFR (Appendix A). This result is in accordance with the previous finding that blood Cd levels increase with age, with small intra-individual variations occurring among those with high Cd exposure [40]. Collectively, these results suggest that age affects the association between Cd levels and eGFR to a certain extent. The multiple regression coefficient of the blood Cd level was −5.473, indicating a non-significant negative relationship with eGFR.

As an environmental vulnerable area, contamination near a mine and its associated health effects have been studied [41,42]. Regarding those efforts, the current study is highly valuable given that both environmental monitoring and biomonitoring were performed in specific areas surrounding an abandoned metal mine. Although there are inherent limitations associated with the use of spot samples, the half-life of heavy metals in the body ranges from weeks to months, making them suitable as exposure indicators. Our results are also strengthened by the inclusion of 26 individuals who had participated in the 2013 study, which allowed us to investigate changes in the heavy metal levels over time. These data may aid in developing appropriate standards for follow-up care in patients with high levels of heavy metal exposure. Notably, the mean age of the participants was 69.4 years, and the mean duration of residence was 45.3 years; thus, our results indicated possible long-term exposure to contaminants near the abandoned metal mine. Given that various adverse events, such as impaired renal function, can occur if the source of exposure is not removed, continuous monitoring of heavy metal levels is necessary to ensure the health and safety of residents near abandoned metal mines.

## 5. Conclusions

In this study, we performed a comprehensive study by simultaneously conducting environmental monitoring and biomonitoring. The results showed that the concentrations of Cd and As in some soil samples exceeded the criteria for concern. In addition, the BCF of Cd in one rice sample was >1.0, indicating a higher accumulation of Cd in agricultural products than in soil. The blood and urinary Cd levels among those living near an abandoned mine were two and four times higher, respectively, than those in the general population in Korea. Furthermore, the Cd levels were higher in those with lower eGFR levels. Heavy metals from abandoned mining areas can accumulate in soil and rice, adversely impacting human health. Continuous environmental monitoring and biomonitoring are required to ensure the safety of the residents. Further studies to evaluate the influence of epigenetic mechanisms on exposure are needed.

## Figures and Tables

**Figure 1 toxics-11-00519-f001:**
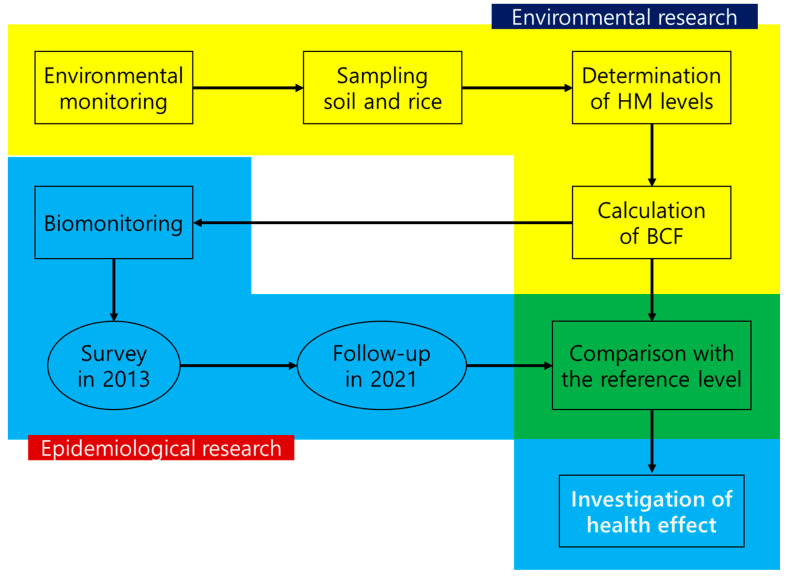
Overall workflow of this study. Abbreviation: HM: heavy metal; BCF: bio-concentration factor.

**Figure 2 toxics-11-00519-f002:**
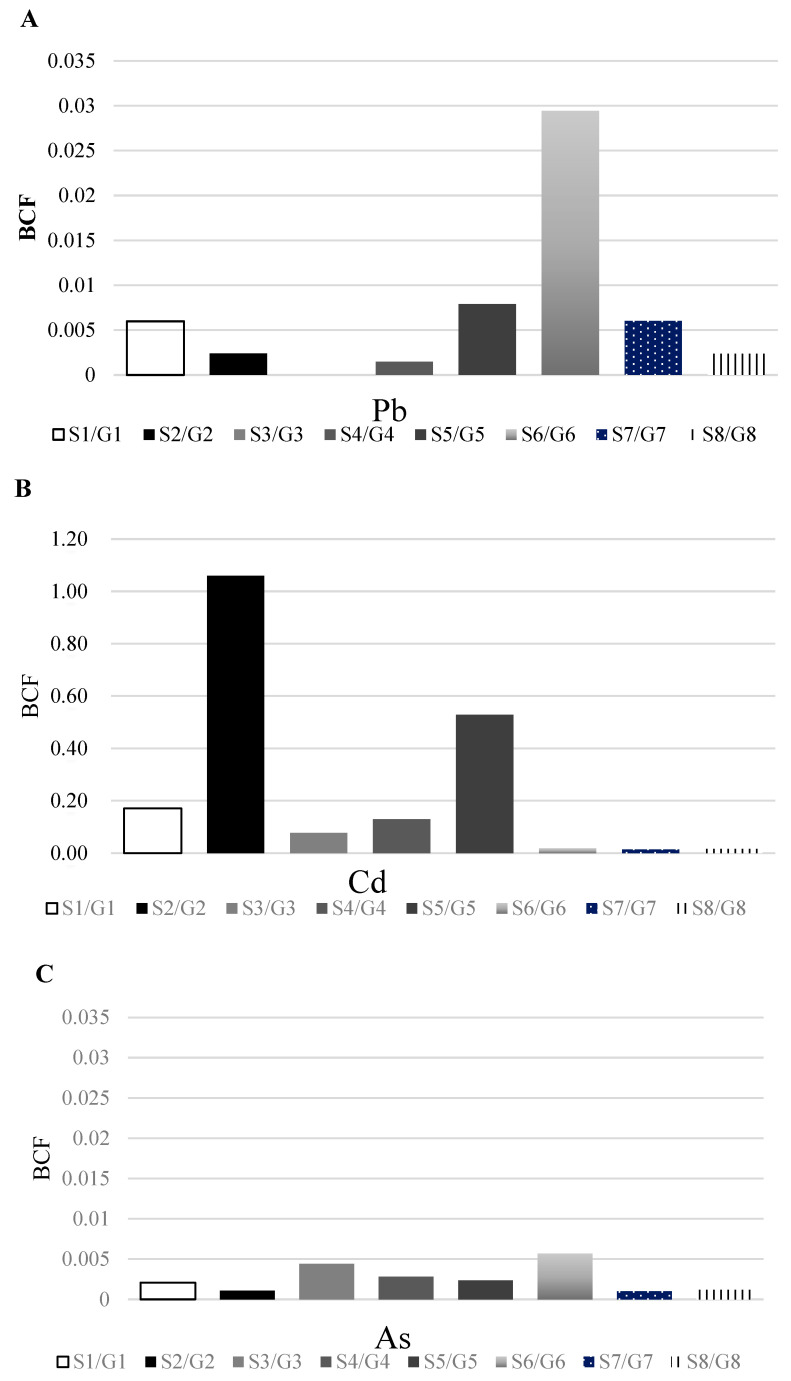
Bio-concentration factors of soil and rice samples: (**A**) lead; (**B**) cadmium; (**C**) arsenic.

**Figure 3 toxics-11-00519-f003:**
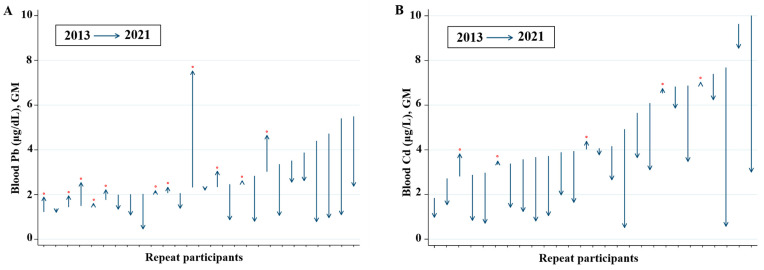
Trends in blood lead and cadmium levels in the study participants. (**A**) Lead; (**B**) Cadmium. The red dot indicates the individuals whose concentration increased in 2021 compared to 2013 and the arrow indicates the direction.

**Figure 4 toxics-11-00519-f004:**
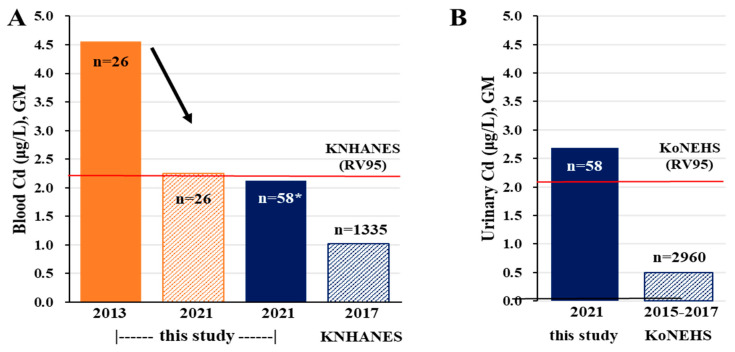
Comparison of cadmium levels with the reference values: (**A**) blood cadmium; (**B**) urinary cadmium. Note: KNHANES (2017) RV95: 2.23 μg/L (among the population aged > 40 years) 3rd KoNEHS RV95: 2.12 μg/L (among the population aged > 40 years). * All participants in this study. Abbreviations: RV95: reference value 95; KNHANES: Korea National Health and Nutrition Examination Survey; KoNEHS: Korea National Environmental Health Survey.

**Table 1 toxics-11-00519-t001:** Concentrations of lead, cadmium, and arsenic in soil and rice samples. Unit: mg/kg.

	Sample	Pb		Criteria	Cd		Criteria	As		Criteria
		Soil	Rice *		Soil	Rice *		Soil	Rice *	
Farm land	S1/G1	25.04	0.15	Concern: 200 Measured: 600	0.82	0.14	Concern: 4 Measured: 12	19.25	0.04	Concern: 25 Measured: 75
	S2/G2	16.61	0.04		1.50	**1.59**		**27.66**	0.03	
	S3/G3	14.40	0.00		1.80	0.14		**27.16**	0.12	
	S4/G4	13.50	0.02		1.16	0.15		**35.14**	0.10	
	S5/G5	12.64	0.10		1.55	**0.82**		**38.18**	0.09	
	S6/G6	14.27	**0.42**		1.11	0.02		24.62	0.14	
	S7/G7	14.97	0.09		0.71	0.01		9.77	0.01	
	S8/G8	12.66	0.03		0.62	0.01		8.38	0.01	
Forest land	S9/-	14.91		Concern: 400 Measured: 1200	**11.09**		Concern: 10 Measured: 30	**182.23**		Concern: 50 Measured: 150
	S10/-	18.24			1.23			63.47		

Note: Criteria for farmland (S1–S8) and forestland (S9, S10) are different. Bold values indicate exceeding the criteria for concern. * In rice, the criterion for concern for all heavy metals is >0.2 mg/kg.

**Table 2 toxics-11-00519-t002:** Clinicodemographic information of the study participants.

	Total	Men	Women	*p*-Value ^^^	
Total *	58 (100)	19 (32.8)	39 (67.2)		
Age (years), AM ± SD	69.4 ± 11.9	67.9 ± 10.2	70.1 ± 12.7	0.513	
BMI (kg/m^2^), AM ± SD	23.8 ± 3.9	23.7 ± 3.5	23.9 ± 4.1	0.810	
Duration of residence (years), AM ± SD	45.3 ± 20.0	49.4 ± 21.1	43.2 ± 19.4	0.272	
Mining work experience *	2 (3.5)	1 (5.3)	1 (2.6)	0.279	
Smoker (past and current) *	12 (20.7)	12 (63.2)	-	<0.001	
Drinker (past and current) *	28 (48.3)	16 (84.2)	12 (30.8)	<0.001	
Local rice consumption >50% *^†^	40 (69.0)	15 (78.9)	25 (64.1)	0.251	
Hypertension *	25 (44.6)	7 (38.9)	18 (47.4)	0.551	
Diabetes *	9 (15.5)	4 (21.1)	5 (12.8)	0.416	
**Metal concentration**, geometric mean (95% confidence interval) unit: μg/L					
Blood Pb (μg/dL)	1.80 (1.54, 2.09)	1.66 (1.25, 2.21)	1.87 (1.55, 2.25)	0.489	
Blood Cd	2.12 (1.73–2.59)	1.21 (0.99–1.50)	2.77 (2.17–3.54)	<0.001	
Urinary Cd	2.69 (2.23–3.25)	2.02 (1.53–2.66)	3.09 (2.43–3.93)	0.022	
Urinary inorganic arsenic (total As sum of As^3+^, As^5+^, MMA, and DMA)	0.22 (0.14, 0.33)	0.27 (0.11, 0.62)	0.20 (0.12, 0.32)	0.519	
Urinary total As	51.8 (42.4, 63.2)	48.8 (36.3, 65.5)	53.3 (40.8, 69.6)	0.706	

Abbreviation: AM: arithmetic mean, SD: standard deviation. * Values are expressed as the number of participants and their proportion. ^ The *p*-values are calculated using the chi-square tests or Wilcoxon tests for the difference between men and women. ^†^ Proportions of locally produced rice consumption to that of total rice consumption.

**Table 3 toxics-11-00519-t003:** General participant characteristics and cadmium levels based on eGFR classification.

Factors	Estimated Glomerular Filtration Rate (mL/min/1.73 m²)			*p*-Value
	<60 (*n* = 2)	60–89 (*n* = 9)	≥90 (*n* = 47)	
Male *	-	3 (33.3)	16 (34.0)	0.603 ^^^
Female *	2 (100)	6 (66.7)	31 (66.0)	
Age (years) ^†^	82 ± 5.7	78.8 ± 9.3	67.0 ± 11.4	0.004
Residence (years) ^†^	68 ± 2.8	49.9 ± 21.7	43.4 ± 19.6	0.093
Smoker (past and current) *	-	2 (22.2)	10 (21.3)	0.762 ^^^
Drinker (past and current) *	-	6 (66.7)	22 (46.8)	0.209 ^^^
Hypertension *	1(50.0)	5 (62.5)	25 (54.4)	0.902 ^^^
Diabetes *	2(100)	-	7 (14.9)	0.002 ^^^
Blood Cd ^#^	7.74 (2.21, 27.09)	2.15 (1.14, 4.07)	1.99 (1.61, 2.48)	0.076
Urine Cd ^#^	3.66 (0.14, 93.53)	3.88 (2.56, 5.89)	2.48 (1.99, 3.07)	0.253

* Values are expressed as the number of participants and their proportion. ^ The *p*-value is calculated using the Chi-square test. ^†^ Values are expressed as the arithmetic mean and their standard deviation (μg/L). ^#^ Values are expressed as the geometric mean and their confidence interval (μg/L).

## Data Availability

The data presented in this study are available on request from the corresponding author. The data are not publicly available due to the ownership belongs to the college of Medicine in Dong-A university.

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
