# Peer review of "Bioaccumulation of Lead, Cadmium, and Arsenic in a Mining Area and Its Associated Health Effects"

_toxics, 2023, doi:10.3390/toxics11060519_

Round 1

Reviewer 1 Report

The authors aimed to investigate bioaccumulation of heavy metals and the associated health impacts based on an integrated assessment of biomonitoring findings and environmental samples obtained near a mining area of the Jinheung Hongcheon mine. It is a very important problem due to huge number of similar sites.

Though the manuscript has been written in a scientific way and Authors applied the proper methods there are few thing which are missing in that work.

The first one - there is no description of the soil sampling points, the information  "All sampling points were located within a 2 km radius of the mine pit." and "while the samples at point S9 in front of the mine shaft exceeded the criterion for concern for the forest area (10 mg/kg), with a level of 11.09 mg/kg." is highly insuficient for proper interpretation of the obtained results. More over it should be "while the sample at point S9 ..."

How big were the soils samples ? Were they the composite ones?

It would be also proper to give bibliography for the pilot study of risk assessment among the residents near abandoned mines since 2012 which were conducted by the Korean Ministry of Environment. Especially, to compare which invertigations were re-conducted. It is not clearly explained.

Lines 142-143 "Standard samples were collected and mixed with urine from the general public, and diluted to 0.05, 0.1, 0.5, 1, and 2 μg/L in a diluted solution respectively, and analyzed by the standard substance addition method" what does it mean - "mixed with urine from the general public"

Did Authors performe any analysis with the control group? Did they use 2017 KNHANES 2017 and 3 rd KoNEHS (2015–2017). Was there no references for 2013 and 2021, as those investigations were provided every 3 years? And by the way - were are the references for them?

It is hard to analyse results because the two groups - inhabitants investigated in 2013 and the remain ones, were put into a one big group. 

In my opinion this survey would be more valuable if Authors would compare both groups instead of treating them as one - it would show the difference between the "old" inhabitants and the "new ones" which did not live in that area for the all time period between both surveys. 

For the future investigations the surveys can be extendet by using human hair as indicators of heavy metals pollution.

Reviewer 2 Report

This paper describes heavy metal contamination levels in the soil around an abandon mine site, and in individuals living nearby.  It also presents a correlative analysis of cadmium lead and arsenic levels in these individuals and glomerular filtration rate.  The study has value as an update on a previous study on heavy-metal levels in the same location.  There are some issues with the manuscript that must be addressed.

1.  Figure 1 abbreviations should be defined in the legend
2. I think it would be useful to the reader to have a simple map of the site, showing the S1-S10 sample site locations relative to the mine and the fields.
3. In figure 3, is there a 1 to 1 correspondence between the individuals in panel A and those in panel B?  For example, does the 4th arrow from the right in panel A represent the same individual as the 4th arrow from the right in panel B?  If so, this this figure should be adjusted in someway to make it easier to compare changes in Cd and Pb levels in the same individuals.  This could be accomplished by labeling each arrow with a letter (a, b, c etc.) or maybe color coding the arrows in some way.  If there is no correspondence this should be clearly indicated.
4. In Figure 4, error bars should be shown.
5. As mentioned in the discussion, the differences in metal concentrations between this study and the previous study conducted in the same area are possibly due to differences in sampling time and location.  I think this is the mostly likely explanation and in any case, the conclusions about temporal trends of toxicant levels at this location should not be made or implied.

English usage and grammar is irregular and occasionally difficulty to understand.  The manuscript should be carefully edited for proper English language usage and writing clarity.

Reviewer 3 Report

The suggestions were made in the attached text.
